# Phased Array Antenna Calibration Based on Autocorrelation Algorithm

**DOI:** 10.3390/s24237496

**Published:** 2024-11-24

**Authors:** Xuan Luong Nguyen, Nguyen Trong Nhan, Thanh Thuy Dang Thi, Tran Van Thanh, Phung Bao Nguyen, Nguyen Duc Trien

**Affiliations:** 1Faculty of Physics, VNU University of Science, Ha Noi 100000, Vietnam; nguyenxuanluong_sdh22@hus.edu.vn (X.L.N.); dangthithanhthuy@vnu.edu.vn (T.T.D.T.); 2Air Defense-Air Force Technical Institute, 166 Hoang Van Thai, Ha Noi 11400, Vietnam; tranthanhtat@gmail.com (T.V.T.); ductrien.minck@gmail.com (N.D.T.); 3Department of Electronic Technology, Institute of System Integration, Le Quy Don Technical University, 236 Hoang Quoc Viet, Ha Noi 11917, Vietnam; nguyenphungbao@lqdtu.edu.vn

**Keywords:** phased array antenna, autocorrelation algorithm, far-field probe, amplitude error, phase error, radiation pattern

## Abstract

The problem of calibrating phased array antennas in a noisy environment using an autocorrelation algorithm is investigated and a mathematical model of the autocorrelation calibration method is presented. The proposed calibration system is based on far-field scanning of the phased array antenna in an environment with internal noise and external interference. The proposed method is applied to a phased array antenna and compared with traditional rotating-element electric-field vector methods, which involve identifying the maximum and minimum vector–sum points (REVmax and REVmin, respectively). The proposed calibration system is verified for a phased array antenna at 3 GHz. Experimental verification of the mathematical model of the proposed method demonstrates that the autocorrelation method is more accurate than the rotating-element electric-field vector methods in determining the amplitude and phase shifts. The measured peak gain of the combined beam in the E-plane increased from 7.83 to 8.37 dB and 3.57 to 4.36 dB compared to the REVmax and REVmin methods, respectively, and the phase error improved from 47° to 55.48° and 19.43° to 29.16°, respectively. The proposed method can be considered an effective solution for large-scale phase calibration at both in-field and in-factory levels, even in the presence of external interference.

## 1. Introduction

Phased array antennas (PAAs) are a pervasive technology in contemporary radio engineering and communication systems [1,2,3,4]. The process of thinning PAAs involves the tuning of antenna elements with uniform spacing to achieve a desired amplitude density across the aperture area. One significant challenge in phased array signal processing is the occurrence of amplitude and phase errors across array channels [5,6,7,8]. This error results in a significant reduction in the performance of the system, particularly with respect to the estimation of the unknown steering vector for the target echo signal and the accuracy of the antenna’s weight vector. Such degradation can result in amplitude–phase distortion, namely, an amplitude–phase discontinuity of the output target echo signal. This can ultimately affect a number of performance aspects, such as anti-jamming capability, accuracy of the digital beamforming, and resolution of synthetic aperture imaging in a variety of applications. In such cases, it is necessary to accurately determine the amplitude and phase shifts of the received signal between channels.

This issue has been addressed through the utilization of an array of calibration techniques, encompassing a calibration line, peripheral fixed probe, mutual coupling, near-field probe, and far-field probe. The calibration line method [9,10,11,12] permits self-calibration through the embedded lines. However, the increase in the requisite number of coupled lines and the potential for error introduced by the coupled line render this method impractical for large-scale PAAs. Additionally, the peripheral fixed-probe method [13,14,15,16,17] employs a probe positioned between the antennas to rectify any discrepancies. Unfortunately, the PAA must be large because of the probe inserted into the array. This method, using a near-field probe antenna [18,19,20,21,22], can detect the amplitude and phase of each element of the PAA through the application of mechanical movement. Still, as the antenna array size increases, the time required for calibration increases significantly because of the mechanical movement of the probe across the entire array. Furthermore, this method is not suitable for in-the-field calibration.

Conversely, the far-field calibration method [23,24,25,26,27,28] entails the measurement of the combined electric-field vector of the PAA based on the reference antenna in the far-field region with the objective of calibrating the magnitude and phase of each element. This method allows for a straightforward calibration of the amplitude and phase of each element based solely on the measurement of the received power. The rotating-element electric-field vector (REV) method entails a rotation of the phase of each antenna element in the PAA from 0° to 360° to identify the maximum and minimum levels of the combined electric-field vectors (REVmax and REVmin, respectively). However, a limitation of this approach is its narrow dynamic range with regard to the change in phase around the maximum and minimum received powers. This results in the system being unable to accurately determine the requisite phase calibration values because of the limited power resolution near the maximum and minimum points of the combined electric-field vectors, resulting in an inherent calibration error.

Furthermore, the efficacy of the calibration processes utilizing these algorithms is significantly compromised by the influence of internal noise and external interference during the operation of PAAs. The objective of a calibration method at an in-field level is to achieve high accuracy because PAAs should exhibit optimal performance and accuracy when mass-produced for use in the field to achieve good reliability. Accordingly, this study puts forth a novel far-field calibration methodology to address the shortcomings of traditional REV techniques. The proposed method determines the phase and amplitude compensation weights subsequent to the calculation of the maximum value of the autocorrelation function (ACF). The proposed algorithm enables the minimization of the impact of complex noise environments during the in-field calibration of PAAs.

The remainder of this paper is configured as follows: Section 2 presents the mathematical model and the simulation approach of the proposed method. Section 3 details the design and measurement of the proposed calibration system followed by a comparison between the proposed method and existing REV methods. The paper is concluded in Section 4.

## 2. Proposed Method: Theory and Simulation

### 2.1. Mathematical Model of the Proposed Method

The fundamental principle of calibration based on an autocorrelation algorithm is to identify the weights that must be applied to compensate for phase and amplitude discrepancies after determining the maximum value of the ACF of the received signal at each receiving channel and the reference signal.

To calibrate the PAA by the autocorrelation algorithm, the following steps are performed:

Step 1. Multiply the received signal by the reference signal.

Step 2. Determine the maximum value of the function at the output of the correlation multiplier.

Step 3. Calculate the weight required to compensate for the input signal.

Step 4. Multiply the received signal by the weight to be compensated.

Step 5. Formulate a signal that combines all the compensated signals in each channel.

The diagram of the PAA calibration method based on the autocorrelation algorithm, as illustrated in Figure 1, comprises the following components: *N* antenna elements and all array elements separated by the same distance *d*, leading to a linear array of total length *D =* (*N* − 1) × *d*; multipliers; a block for determining $w_{i}$, the maximum value of the function at the output of the correlation multiplier; a block for calculating $a_{i}$ (the weight needed to compensate for the input signal), which permits the determination of the amplitude and phase compensation; and an adder.

In the following steps, we derive a mathematical formula that represents the calibration process of a PAA in a noisy environment.

1.Receive signals in channels of the following form:

(1)$$x_{i}\left(t\right)=s_{i}\left(t\right)+n_{i}\left(t\right),$$ where $s_{i}\left(t\right)=A_{i}e^{j\beta_{i}\left(t\right)}$ is the complex useful signal obtained at the *i*-th element of the array antenna, *i* = 1, 2,..., *N*, at time *t*, where $A_{i}$ and $\beta_{i}\left(t\right)$ are the amplitude and phase of the useful signal obtained, respectively, and $n_{i}\left(t\right)$ is the corresponding noise, which includes internal noise and external interference.

2.Multiply the received signal $x_{i}\left(t\right)$ by the reference signal $x_{0}\left(t\right)$. The signal at the output of the multiplier has the following form:

(2)$$\begin{array}{c} X_{i}\left(t\right)=x_{i}\left(t\right)\times x_{0}\left(t\right)=\left(s_{i}\left(t\right)+n_{i}\left(t\right)\right)\times \left(s_{0}\left(t\right)+n_{0}\left(t\right)\right)\\ \;\;\;\;\;\;=s_{i}\left(t\right)\times s_{0}\left(t\right)+s_{i}\left(t\right)\times n_{0}\left(t\right)+n_{i}\left(t\right)\times s_{0}\left(t\right)\\ \;\;\;\;\;\;+n_{i}\left(t\right)\times n_{0}\left(t\right), \end{array}$$
where $x_{0}\left(t\right)=s_{0}\left(t\right)+n_{0}\left(t\right)$ and $s_{0}\left(t\right)=A_{0}e^{j\beta_{0}\left(t\right)}$ are the complex reference signal at time *t*, where $A_{0}$ and $\beta_{0}\left(t\right)$ are the amplitude and phase of the reference signal, respectively, and $n_{0}\left(t\right)$ is the corresponding noise, which exclusively pertains to the internal noise generated during the signal generation and transmission process to the multipliers. In the case of white noise, we know that the correlation function of white noise has the following form [29]: $R\left(t\right)=\frac{C_{0}}{2\pi }\int_{-\infty }^{\infty }e^{j\omega t}d\omega =C_{0}\delta \left(t\right)$, where $C_{0}=C\left(\omega \right)=\mathrm{constant}$ and R(t) is zero at all points except t=0.

3.Find the maximum value of the ACF $X_{i}\left(t\right)$ in the block to determine $w_{i}$:

(3)$$\begin{array}{c} w_{i}=\max(X_{i}\left(t\right))=\max\left(x_{i}\left(t\right)\times x_{0}\left(t\right)\right)\\ \;\;\;\;\;\;=\max(s_{i}\left(t\right)\times s_{0}\left(t\right)+s_{i}\left(t\right)\times n_{0}\left(t\right)+n_{i}\left(t\right)\times s_{0}\left(t\right)\\ \;\;\;\;\;\;+n_{i}\left(t\right)\times n_{0}\left(t\right)), \end{array}$$
where $X_{i}\left(t\right)$ denotes the ACF of the received signal $x_{i}\left(t\right)$ and the reference signal $x_{0}\left(t\right)$.

4.Calculate the weight $a_{i}$ needed to compensate for the input signal if channel *k* (*k* = 1, 2, …, *N*) is taken as the standard according to the following formula:



(4)
$$\begin{array}{c} a_{i}=\frac{w_{k}}{w_{i}}=\frac{\max\left(X_{k}\left(t\right)\right)}{\max\left(X_{i}\left(t\right)\right)}=\frac{\max\left(x_{k}\left(t\right)\times x_{0}\left(t\right)\right)}{\max\left(x_{i}\left(t\right)\times x_{0}\left(t\right)\right)}\\ =\frac{\max\left(s_{k}\left(t\right)\times s_{0}\left(t\right)+s_{k}\left(t\right)\times n_{0}\left(t\right)+n_{k}\left(t\right)\times s_{0}\left(t\right)+n_{k}\left(t\right)\times n_{0}\left(t\right)\right)}{\max\left(s_{i}\left(t\right)\times s_{0}\left(t\right)+s_{i}\left(t\right)*n_{0}\left(t\right)+n_{i}\left(t\right)\times s_{0}\left(t\right)+n_{i}\left(t\right)\times n_{0}\left(t\right)\right)}. \end{array}$$



When the *k*th receiver channel is the standard channel, then $a_{k}=1$.

5.Multiply the received signal $x_{i}\left(t\right)$ by the weight to be compensated $a_{i}$:



(5)
$$\begin{array}{c} Y_{i}\left(t\right)=x_{i}\left(t\right)\times a_{i}\\ =\left(s_{i}\left(t\right)+n_{i}\left(t\right)\right)\\ \times \frac{\max\left(s_{k}\left(t\right)\times s_{0}\left(t\right)+s_{k}\left(t\right)\times n_{0}\left(t\right)+n_{k}\left(t\right)\times s_{0}\left(t\right)+n_{k}\left(t\right)\times n_{0}\left(t\right)\right)}{\max\left(s_{i}\left(t\right)\times s_{0}\left(t\right)+s_{i}\left(t\right)\times n_{0}\left(t\right)+n_{i}\left(t\right)\times s_{0}\left(t\right)+n_{i}\left(t\right)\times n_{0}\left(t\right)\right)}. \end{array}$$



When the *k*th receiver channel is the standard channel, then $Y_{k}\left(t\right)=x_{k}\left(t\right)$.

6.The array coefficient function at the adder output is written as follows:


(6)
$$\begin{array}{c} AF_{i}=Y_{1}\left(t\right)+Y_{2}\left(t\right)+Y_{3}\left(t\right)+\ldots +Y_{N}\left(t\right)=\overset{N}{\underset{i=1}{\sum }}Y_{i}\left(t\right)\\ =\overset{N}{\underset{i=1}{\sum }}\frac{\left(s_{i}\left(t\right)+n_{i}\left(t\right)\right)\times \max\left(s_{k}\left(t\right)\times s_{0}\left(t\right)+s_{k}\left(t\right)\times n_{0}\left(t\right)+n_{k}\left(t\right)\times s_{0}\left(t\right)+n_{k}\left(t\right)\times n_{0}\left(t\right)\right)}{\max\left(s_{i}\left(t\right)\times s_{0}\left(t\right)+s_{i}\left(t\right)\times n_{0}\left(t\right)+n_{i}\left(t\right)\times s_{0}\left(t\right)+n_{i}\left(t\right)\times n_{0}\left(t\right)\right)}. \end{array}$$


When the first receiver channel is the standard channel, the array coefficient function $AF_{i}$ output is written as follows:(7)$$\begin{array}{c} AF_{i}=x_{1}\left(t\right)+Y_{2}\left(t\right)+Y_{3}\left(t\right)+\ldots +Y_{N}\left(t\right)=1+\overset{N}{\underset{i=2}{\sum }}Y_{i}\left(t\right)\\ \;\;\;\;\;=x_{1}\left(t\right)+\overset{N}{\underset{i=2}{\sum }}\frac{\left(s_{i}\left(t\right)+n_{i}\left(t\right)\right)\times \max\left(s_{i}\left(t\right)\times s_{0}\left(t\right)+s_{i}\left(t\right)\times n_{0}\left(t\right)+n_{i}\left(t\right)\times s_{0}\left(t\right)+n_{i}\left(t\right)\times n_{0}\left(t\right)\right)}{\max\left(s_{i}\left(t\right)\times s_{0}\left(t\right)+s_{i}\left(t\right)\times n_{0}\left(t\right)+n_{i}\left(t\right)\times s_{0}\left(t\right)+n_{i}\left(t\right)\times n_{0}\left(t\right)\right)}. \end{array}$$

### 2.2. Simulation of the Proposed Method

The efficiency of the PAA calibration is evaluated based on the mathematical model of the autocorrelation algorithm, developed in Section 2.1, through simulation in a MATLAB environment.

In the case where *N* = 2 and receiver channel 1 ($a_{1}=1$) is the reference, the weight $a_{2}$ needed to compensate for the input signal of receiver channel 2 is expressed as follows:(8)$$\begin{array}{c} a_{2}=\frac{w_{1}}{w_{2}}=\frac{\max(X_{1}\left(t\right))}{\max(X_{2}\left(t\right))}=\frac{\max\left(x_{1}\left(t\right)\times x_{0}\left(t\right)\right)}{\max\left(x_{2}\left(t\right)\times x_{0}\left(t\right)\right)}\\ =\frac{\max\left(s_{1}\left(t\right)\times s_{0}\left(t\right)+s_{1}\left(t\right)\times n_{0}\left(t\right)+n_{1}\left(t\right)\times s_{0}\left(t\right)+n_{1}\left(t\right)\times n_{0}\left(t\right)\right)}{\max\left(s_{2}\left(t\right)\times s_{0}\left(t\right)+s_{2}\left(t\right)\times n_{0}\left(t\right)+n_{2}\left(t\right)\times s_{0}\left(t\right)+n_{2}\left(t\right)\times n_{0}\left(t\right)\right)}. \end{array}$$

Then, for the first channel, $Y_{1}\left(t\right)=x_{1}\left(t\right)$.

The received channel 2 signal, after being multiplied by the compensation weight, $a_{2}$ has the following form:(9)$$\begin{array}{c} Y_{2}\left(t\right)=x_{2}\left(t\right)\times a_{2}\\ =\left(s_{2}\left(t\right)+n_{2}\left(t\right)\right)\\ \times \frac{\max\left(s_{1}\left(t\right)\times s_{0}\left(t\right)+s_{1}\left(t\right)\times n_{0}\left(t\right)+n_{1}\left(t\right)\times s_{0}\left(t\right)+n_{1}\left(t\right)\times n_{0}\left(t\right)\right)}{\max\left(s_{2}\left(t\right)\times s_{0}\left(t\right)+s_{2}\left(t\right)\times n_{0}\left(t\right)+n_{2}\left(t\right)\times s_{0}\left(t\right)+n_{2}\left(t\right)\times n_{0}\left(t\right)\right)}. \end{array}$$

Then, the array coefficient function at the adder output is written as follows:(10)$$AF_{2}=Y_{1}\left(t\right)+Y_{2}\left(t\right)=x_{1}\left(t\right)+\frac{\left(s_{2}\left(t\right)+n_{2}\left(t\right)\right)\times \max\left(s_{1}\left(t\right)\times s_{0}\left(t\right)+s_{1}\left(t\right)\times n_{0}\left(t\right)+n_{1}\left(t\right)\times s_{0}\left(t\right)+n_{1}\left(t\right)\times n_{0}\left(t\right)\right)}{\max\left(s_{2}\left(t\right)\times s_{0}\left(t\right)+s_{2}\left(t\right)\times n_{0}\left(t\right)+n_{2}\left(t\right)\times s_{0}\left(t\right)+n_{2}\left(t\right)\times n_{0}\left(t\right)\right)}.$$

The subsequent step involves the simulation of an exemplary calibration scenario, illustrating the proposed method. The reference antenna, the antennas to be calibrated, and the interference source are positioned at specified distances from one another within the far-field region, as illustrated in Figure 2. The reference antenna is aligned toward the middle of the two antennas to be calibrated to accurately detect amplitude and phase shifts. The interference source is positioned in front of the two antennas to be calibrated. In this step, receiving antenna 1 (green) and receiving antenna 2 (light blue) are calibrated according to the reference antenna (orange) using the proposed method.

To verify the reliability of the proposed calibration system, 10 measurements were repeated. Table 1 and Table 2 comprehensively present the simulation results for the amplitude and phase errors when *D* = 0.625λ = 0.0625 m for the case λ = 0.1 m. The autocorrelation method for the PAA exhibited superior performance compared to the REV methods. Table 1 illustrates that the amplitude error improved by 0.4 and 0.19 compared to the REVmax and REVmin methods, respectively. Table 2 demonstrates that the phase error improved by 42.24° and 16.72° compared to the REVmax and REVmin methods.

Figure 3 demonstrates the simulated radiation patterns for the case *N* = 2, *D* = 0.625*λ* and azimuth angle of the transceiver antenna *φ_tr._* = 90° when using the REVmax, REVmin, and autocorrelation methods. The results of the simulation demonstrate that the autocorrelation method exhibits superior amplitude and phase calibration accuracy compared to the REVmax and REVmin methods. The results illustrate that the measured peak gain in the E-plane improved by 8 dB and 4 dB compared to the REVmax and REVmin methods, respectively. In addition, without an interference source after calibration, the beams were tilted by 1°, 5°, and 10° for the autocorrelation, REVmin, and REVmax methods.

Consequently, the autocorrelation method has certain advantages over the REV methods. In the case of an array of *N* elements (*N* > 2), the AFC method continues until the final sequence is reached. Once an initial pair of elements is selected for comparison and calibration, one of the two elements is retained as the reference element to contrast with the subsequent element. This approach also serves to minimize phase variation among the phase shifters, attributable to the intrinsic nature of the active component. The autocorrelation method facilitates the precise determination of the amplitude and phase shifts, enabling the calibration of large-scale PAAs to achieve the maximum combined beam peak after calibration in the presence of external noise affecting performance. In contrast, the REV methods yield inaccurate amplitude and phase shifts for the two initial elements, which can result in erroneous amplitude and phase shifts for subsequent elements. This can lead to a reduction in the amplitude of the combined beam at the output combiner, which affects the efficiency of subsequent signal processing.

## 3. Results and Discussion

To empirically validate the mathematical model based on the autocorrelation algorithm, a series of experiments using a calibration system design was conducted. The amplitude and phase errors of the proposed array with varying signal–to–noise ratios at the input of the receiver, and the radiation patterns of the proposed array following the implementation of amplitude and phase shifts on received signals when employing various methods were compared. Subsequently, the design of the experimental calibration system, used to perform the aforementioned tasks, is described in detail.

### 3.1. Proposed Calibration System Design

Figure 4 depicts the calibration system used to determine the amplitude and phase errors of the proposed array when using the various methods. The system comprised the following components: a laptop with MATLAB 2024a; a signal processing block (Figure 5a), comprising an AD9361 card and a ZC706 Xilinx FPGA board; a reference antenna (depicted in Figure 5b); two calibration receiving antennas (Figure 5c); and an interference source. The reference antenna, calibration receiving antennas, and interference source are distanced from each other in the far-field region, as shown in Figure 4. The reference antenna is oriented in a central position relative to the two antennas to be calibrated. The phased array calibration process is described as follows: A reference antenna is used to transmit the calibration signal. Subsequently, the signal received from the two receiving antennas is digitized using the AD9361 card, which is fabricated on an Analog Devices in United States. The signal from the AD9361 card is transferred to the ZC706 board to determine the amplitude and phase shifts on the laptop when using the REVmax, REVmin, and autocorrelation methods. The entire calibration system is controlled through a graphical user interface, as illustrated in Figure 6.

To verify the reliability of the proposed method when using this calibration system, a total of 10 trials were repeated. The following initial parameter values were set: the distance between two calibration receiving antennas was *D* = 0.625*λ* and *D* = 1.25*λ*; the distance between the PAA under calibration and the reference antenna was 3 m, which satisfied the far-field criteria at 3 GHz; the distance between the reference antenna and the interference source was 2 m; the distance between the PAA under calibration and the interference source antenna was 4 m; the interference power was equal to 10 dBm; and the interference spectrum bandwidth was equal to 20 MHz. It is worth noting the REVmax and REVmin approaches could also be implemented using the proposed calibration system.

Figure 7 illustrates the calibration system for displaying the radiation patterns of the proposed array following the implementation of the amplitude and phase shifts on the received signals using the three methods. The system comprises the following components: a laptop with MATLAB 2024a; a signal processing block; two calibration receiving antennas; and one transmitting antenna (depicted in Figure 5c). The transceiver antenna and calibration antennas are distanced from each other in the far-field region, as shown in Figure 6. The transceiver antenna is aligned with the two calibration receiving antennas. In the initial stage of the process, a transceiver antenna is employed to transmit the signal. Subsequently, the signal received by the two receiving antennas is digitized using the AD9361 card. The signal from the AD9361 card is transferred to the ZC706 board to produce the radiation pattern of the received signals on the laptop when using the REVmax, REVmin, and autocorrelation methods. Furthermore, a graphical user interface for automated calibration control was developed using MATLAB, as illustrated in Figure 6.

### 3.2. Measurement and Comments

Subsequently, the amplitude and phase of the PAAs were calibrated using the proposed calibration system in Section 3.1. Table 3, Table 4, Table 5 and Table 6 present a comprehensive account of the experimental outcomes pertaining to the measured amplitude and phase errors. The autocorrelation method in the PAAs exhibited superior performance compared to the REV methods. Table 3 illustrates that when *D* = 0.625λ, the amplitude error improved by 0.45 and 0.19 compared to the REVmax and REVmin methods, respectively. Table 4 shows that the phase error improved by 47° and 19.43° compared to the REVmax and REVmin methods when *D* = 0.625λ. Table 5 illustrates that when *D* = 1.25λ, the amplitude error improved by 0.49 and 0.22 compared to the REVmax and REVmin methods. Table 6 shows that the phase error improved by 55.48° and 29.16° compared to the REVmax and REVmin methods when *D* = 1.25λ.

Figure 8 illustrates the calibration outcomes when employing the REVmax, REVmin, and autocorrelation methods. Figure 8a–d illustrate the radiation patterns of the combined beam when employing distinct calibration techniques where the distance *D* = 0.625λ and azimuth angle of the transceiver antenna is *φ_tr._* = 0°; *D* = 1.25λ and *φ_tr._* = 0°; *D* = 0.625λ and *φ_tr_^.^* = 240°; and *D* = 1.25λ and *φ_tr._* = 240°, respectively. Doubling the distance *D* between elements of the array changed the peak and width of the side lobes. Alterations in the angle *φ_tr_* did not markedly influence the precision of the calibration process, at least within the context of the present investigation. The autocorrelation method in PAAs demonstrated superior performance when compared to the REV methods. Table 7 illustrates that the measured peak gain in the E-plane improved by 7.83 dB and 3.57 dB compared to the REVmax and REVmin methods, respectively. Table 8 shows the measured peak gain in the E-plane improved by 8.37 dB and 4.36 dB when compared to the REVmax and REVmin methods. In addition, without an interference source after calibration, the beams at *D* = 0.625λ were tilted by 1°, 8°, and 14° for the autocorrelation, REVmin, and REVmax methods; for *D* = 1.25λ, the beams were tilted by 2°, 12°, and 16°, respectively.

The radiation pattern of the combined beam is subject to significant alteration when different noises are introduced. These undesired amplitude and phase shifts can be attributed to several factors associated with the RF hardware, including antennas, phase shifters, attenuators, amplifiers, switches, connectors, transmission lines, coaxial cables, and waveguides. Of note, any RF component use across all element channels cannot be the source of relative phase and amplitude shifts. The phase of the received signal is contingent upon not only the shift induced by the receiver channel, but also the phase difference between the probe and the local oscillator signal. Nevertheless, the autocorrelation method still ensures a superior radiation pattern to the REV methods.

## 4. Conclusions

The excitation of a phased array element (in terms of both amplitude and phase) in a noisy environment inevitably deviates from the ideal values, which would otherwise result in degradation of the array’s performance. Therefore, careful calibration and compensation are essential to achieve optimal results when designing a practical phased array system. This work puts forth a novel far-field calibration system with the potential to enhance accuracy and reduce system complexity.

The proposed method is distinguished from existing far-field-based solutions through the use of an autocorrelation algorithm. In contrast, conventional methods based on the REV approach track the maximum and minimum magnitudes of two vector–sum elements in the array. Subsequently, the proposed method is validated through the implementation of a calibration system at 3 GHz and its performance benchmarked against that of conventional REV methods. To ascertain the accuracy of the proposed system, 10 trials were conducted. The simulated results, obtained with the proposed method, were found to align with the measurement results obtained using the far-field measurement method. A comparison of the far-field measured results reveals that the proposed method is both feasible and efficient. Therefore, the proposed method can be considered an effective solution for large-scale phase calibration in both in-field and in-factory settings, even in the presence of external noise.

It should be noted that in the case of a non-uniform arrangement of the PAA elements, the proposed method requires a complex calibration algorithm, which leads to a decrease in the calibration accuracy. Therefore, the authors’ plans include improving the signal calibration algorithm under interference conditions for PAAs with non-uniform arrangements of antenna elements as well as the study of signal calibration characteristics under these conditions.

## Figures and Tables

**Figure 1 sensors-24-07496-f001:**
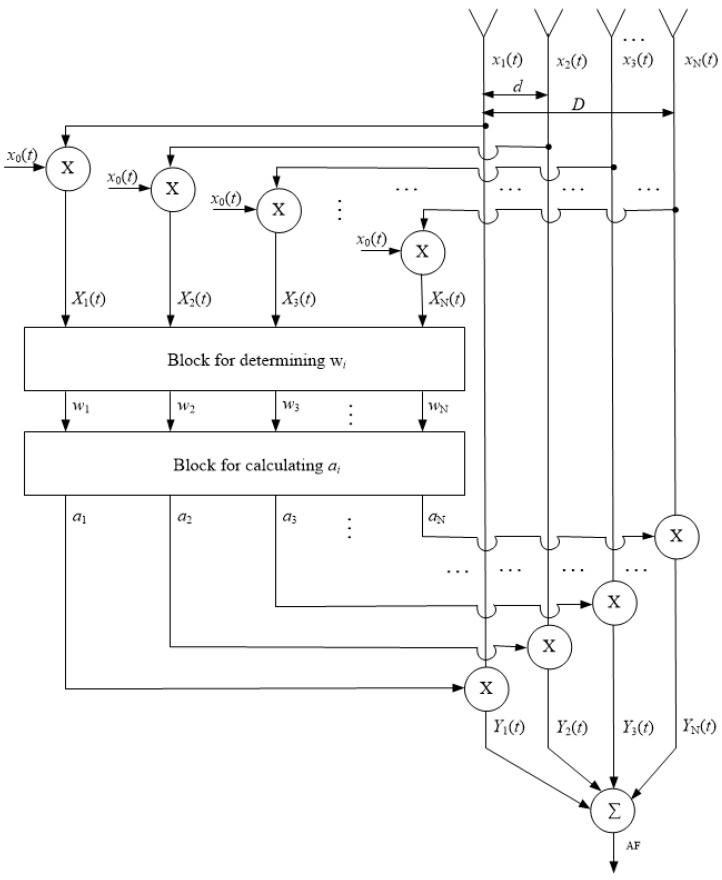
A Schematic diagram of the PAA calibration method based on the autocorrelation algorithm.

**Figure 2 sensors-24-07496-f002:**
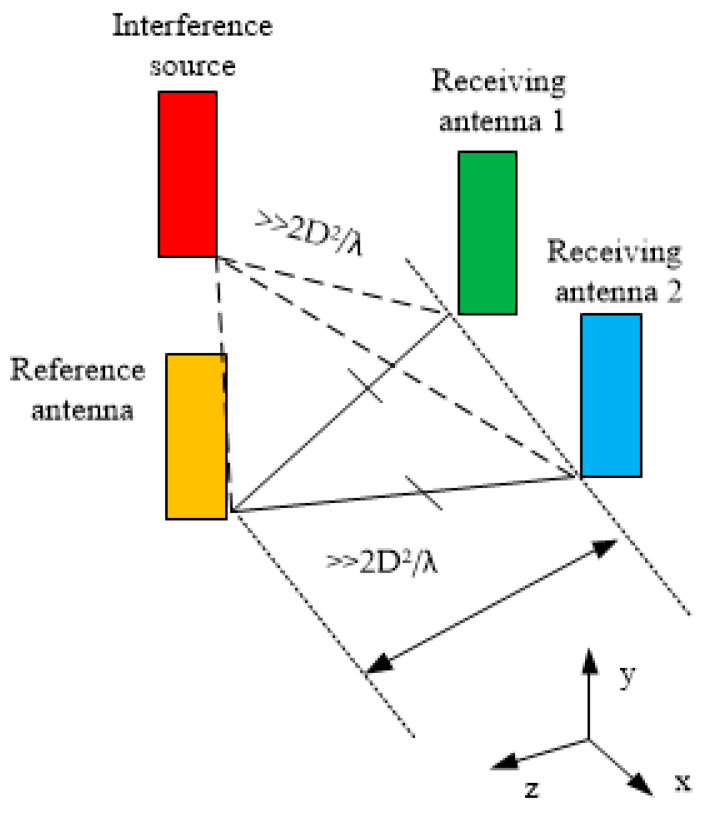
The simulation environment of the calibration showing reference (orange), fixed (green), and calibrated (blue) antennas and interference source (red): 2D^2^/λ = far-field distance in meters; λ = wavelength of the radiating wave in meters.

**Figure 3 sensors-24-07496-f003:**
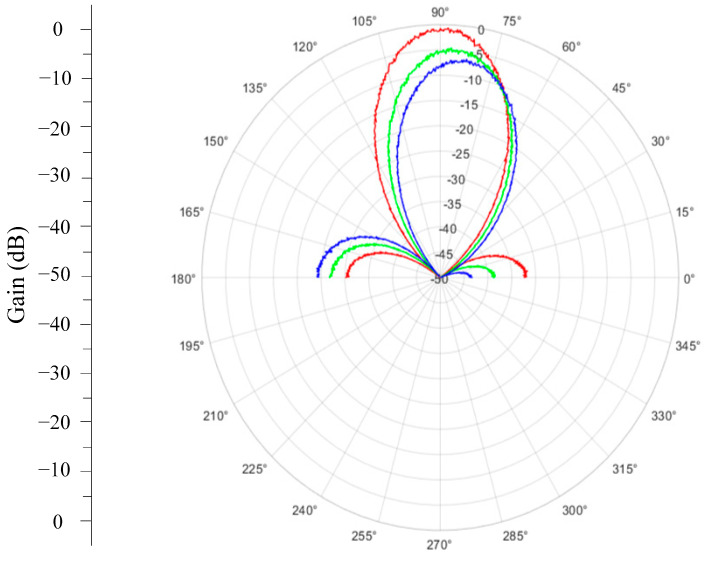
The radiation patterns of the combined beam in the E-plane when using different calibration methods: autocorrelation method—red; REVmin method—green; REVmax method—blue.

**Figure 4 sensors-24-07496-f004:**
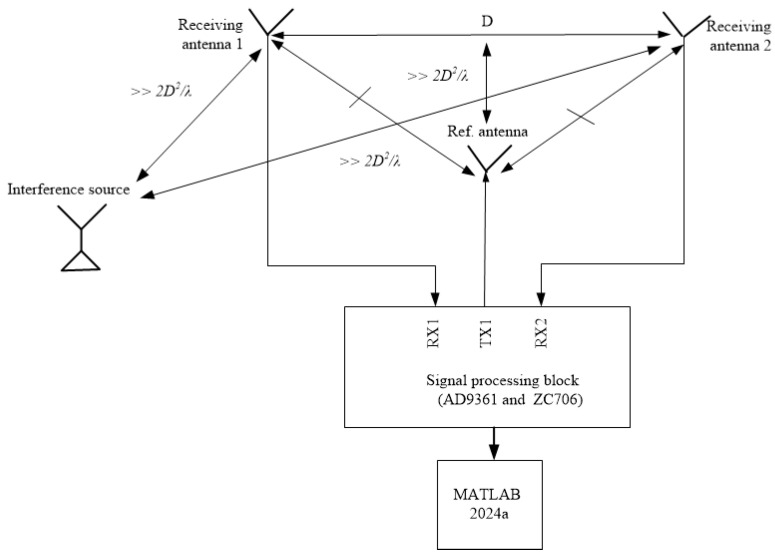
The calibration system diagram for the REVmax, REVmin, and autocorrelation methods.

**Figure 5 sensors-24-07496-f005:**
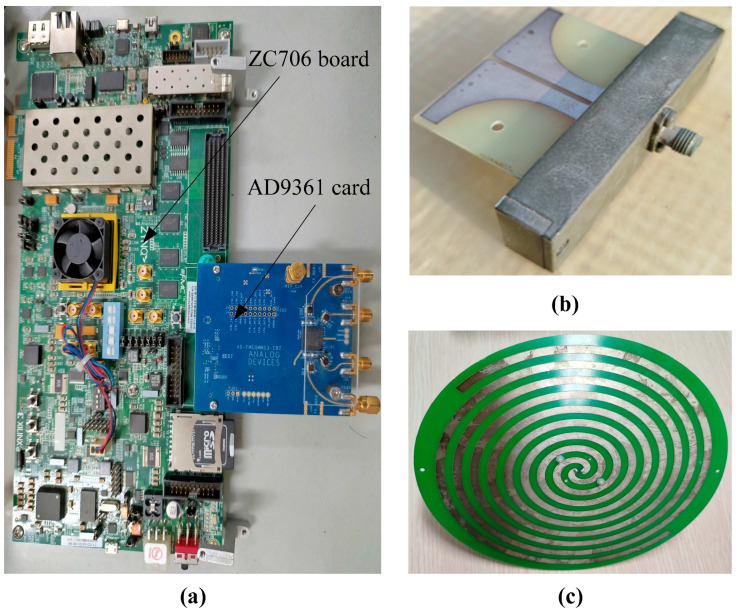
The (**a**) signal processing block, (**b**) receiving antenna, and (**c**) reference (transmitting) antenna.

**Figure 6 sensors-24-07496-f006:**
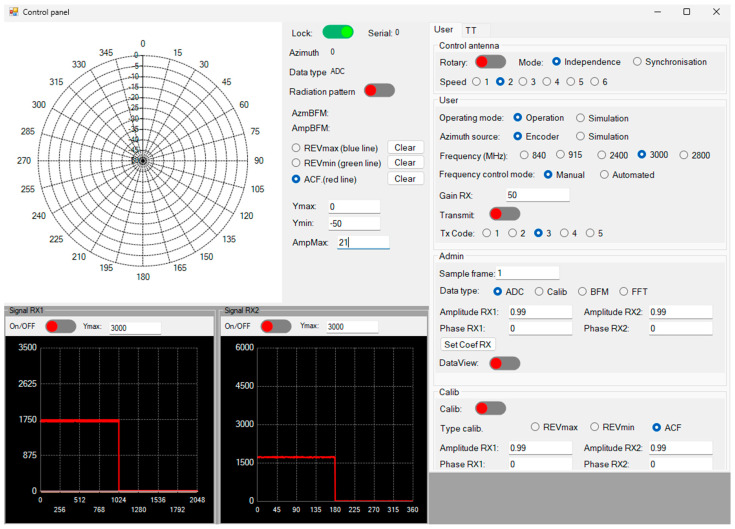
The software graphical user interface for automatic calibration control.

**Figure 7 sensors-24-07496-f007:**
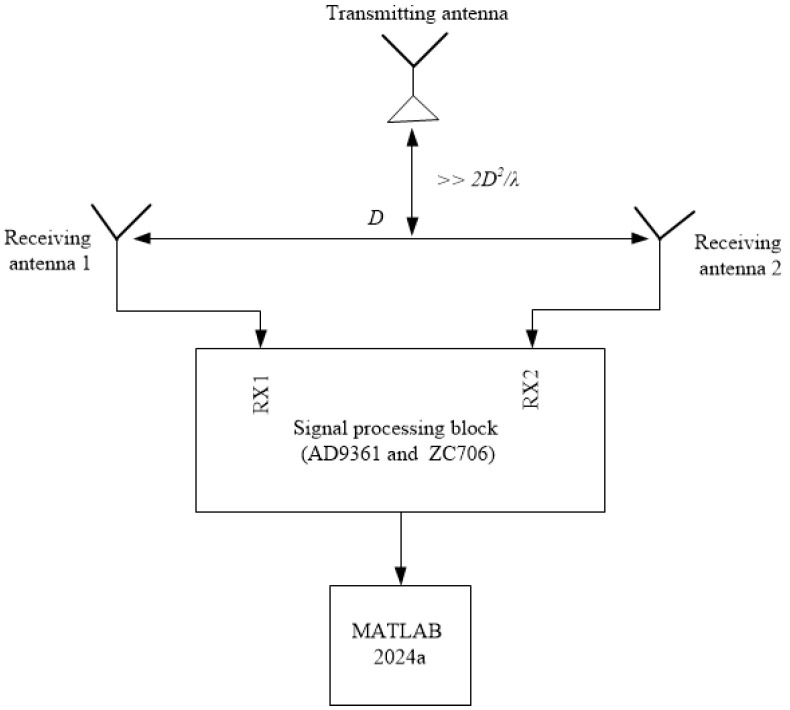
A diagram of the calibration system that produced the radiation patterns after performing the amplitude and phase shifts of the received signals using the REVmax, REVmin, and autocorrelation methods.

**Figure 8 sensors-24-07496-f008:**
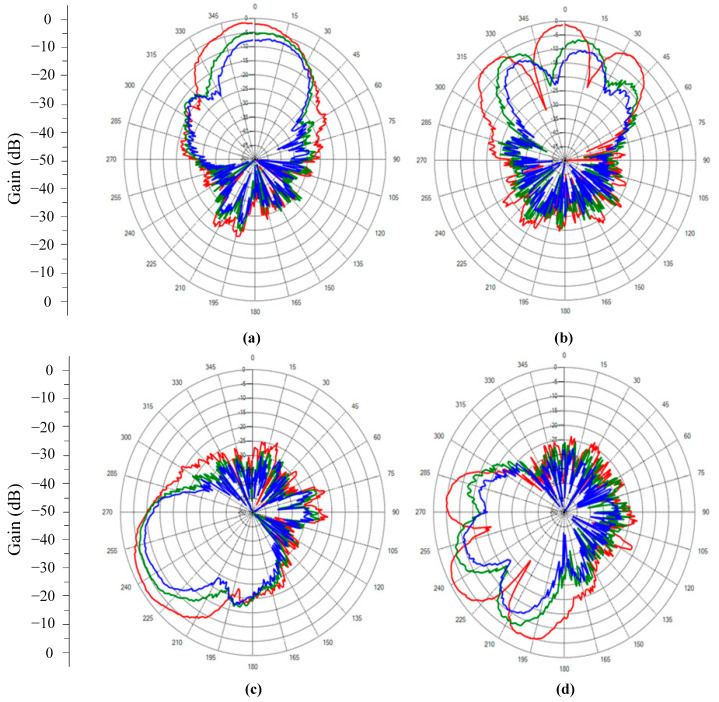
The radiation patterns of the combined beam in the E-plane when using different calibration methods in the cases (**a**) D=0.625λ and *φ_tr_.* = 0°; (**b**) D=1.25λ and *φ_tr_.* = 0°; (**c**) D=0.625λ and *φ_tr_.* = 240°; and (**d**) D=1.25λ and *φ_tr._* = 240°. Autocorrelation method—red; REVmin method—green; REVmax method—blue.

**Table 1 sensors-24-07496-t001:** The amplitude errors (dB) in the calibration results.

Measurement No.		1	2	3	4	5	6	7	8	9	10
Method	REVmax	0.51	0.65	0.62	0.50	0.52	0.67	0.61	0.53	0.55	0.70
	REVmin	0.39	0.37	0.44	0.32	0.37	0.40	0.45	0.30	0.36	0.41
	ACF	0.13	0.15	0.17	0.25	0.10	0.20	0.27	0.25	0.21	0.17

**Table 2 sensors-24-07496-t002:** The phase errors (°) in the calibration results.

Measurement No.		1	2	3	4	5	6	7	8	9	10
Method	REVmax	50.10	38.25	52.74	45.12	39.35	37.50	51.60	44.36	42.89	55.03
	REVmin	25.15	17.15	24.69	17.36	13.57	19.20	8.64	25.74	27.04	23.13
	ACF	4.20	5.06	3.26	1.19	3.56	2.98	2.10	5.10	4.67	2.38

**Table 3 sensors-24-07496-t003:** The amplitude error (dB) of the calibration results when *D* = 0.625λ.

Measurement No.		1	2	3	4	5	6	7	8	9	10
Method	REVmax	0.55	0.71	0.68	0.58	0.62	0.76	0.66	0.59	0.57	0.73
	REVmin	0.42	0.34	0.36	0.42	0.36	0.35	0.49	0.36	0.34	0.45
	ACF	0.23	0.16	0.15	0.30	0.11	0.23	0.31	0.21	0.16	0.13

**Table 4 sensors-24-07496-t004:** The phase error (°) of the calibration results when *D* = 0.625λ.

Measurement No.		1	2	3	4	5	6	7	8	9	10
Method	REVmax	54.16	43.15	59.78	50.20	46.38	42.25	57.26	40.71	48.76	60.50
	REVmin	30.02	14.25	28.10	21.28	19.62	16.19	10.08	25.91	33.52	28.62
	ACF	5.10	2.06	1.18	4.46	2.11	3.63	1.22	4.46	5.31	3.73

**Table 5 sensors-24-07496-t005:** The amplitude error (dB) of the calibration results when *D* = 1.25λ.

Measurement No.		1	2	3	4	5	6	7	8	9	10
Method	REVmax	0.84	0.68	0.57	0.56	0.76	0.85	0.78	0.87	0.69	0.88
	REVmin	0.36	0.65	0.46	0.45	0.59	0.52	0.45	0.36	0.54	0.44
	ACF	0.11	0.25	0.19	0.28	0.44	0.39	0.22	0.15	0.32	0.23

**Table 6 sensors-24-07496-t006:** The phase error (°) of the calibration results when *D* = 1.25λ.

Measurement No.		1	2	3	4	5	6	7	8	9	10
Method	REVmax	66.70	52.11	68.00	60.51	55.32	54.52	65.24	50.20	57.40	70.15
	REVmin	40.54	25.26	38.29	30.62	28.30	35.69	20.10	36.36	42.69	39.10
	ACF	6.32	1.15	7.05	5.10	3.64	5.24	4.36	2.12	3.34	7.05

**Table 7 sensors-24-07496-t007:** The measured peak gains (dB) in the E-plane of the radiation patterns of the combined beam after calibration methods when *D* = 0.625λ.

Measurement No.		1	2	3	4	5	6	7	8	9	10
Method	REVmax	−9.45	−9.22	−10.15	−10.20	−9.62	−9.83	−9.23	−9.81	−9.52	−9.34
	REVmin	−5.11	−5.00	−5.68	−5.62	−5.41	−5.23	−5.82	−5.00	−5.26	−5.68
	ACF	−1.48	−1.53	−1.99	−1.94	−1.57	−1.52	−2.04	−1.82	−2.14	−2.06

**Table 8 sensors-24-07496-t008:** The measured peak gains (dB) in the E-plane of the radiation patterns of the combined beam after calibration methods when *D* = 1.25λ.

Measurement No.		1	2	3	4	5	6	7	8	9	10
Method	REVmax	−10.82	−9.84	−10.93	−9.92	−11.34	−10.93	−12.41	−9.89	−9.56	−12.03
	REVmin	−5.93	−6.55	−7.45	−6.73	−7.65	−6.81	−7.78	−5.03	−6.34	−7.32
	ACF	−1.21	−2.06	−3.24	−2.02	−3.16	−2.72	−3.12	−1.51	−1.87	−3.05

## Data Availability

The original contributions presented in this study are included in the article; further inquiries can be directed to the corresponding author.

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
