# Peer review of "Phased Array Antenna Calibration Based on Autocorrelation Algorithm"

_sensors, 2024, doi:10.3390/s24237496_

Round 1

Reviewer 1 Report

Comments and Suggestions for Authors

In this paper, the effect of amplitude and phase errors between array channels on the performance of antenna systems is investigated and a mathematical model for the autocorrelation calibration method is proposed. The improvements suggested in this paper are as follows:

1. In the introduction, the authors mention the mutual coupling method [18-22], “in the case of a non-uniform arrangement of antenna elements, this method necessitates the use of a complex calibration algorithm, which in turn results in a reduction in calibration accuracy”, for the autocorrelation calibration method in this paper in the case of a non-uniform arrangement of antenna elements, this method necessitates the use of a complex calibration algorithm, which in turn results in a reduction in calibration accuracy? The authors also mentioned that the time required for calibration of near-field probes [23-27] increases significantly as the size of the array antenna increases, will the adaptive calibration proposed in this paper also be affected by this? Because the authors only studied the high precision adaptive calibration for amplitude and phase in the whole paper, they did not analyze whether the method of this paper has any effect on the uneven arrangement of the antenna elements and the variation of the size. But in the references it is mentioned that other methods can have these problems.

2. Section 2.1 mentions “We find a mathematical formula that represents the calibration process of a phased array antenna in a noisy environment”.Is the formula cited, and if so, please cite the reference to the formula.

3. In equations (1) to (7), it is recommended that all variables be defined at first occurrence, and in particular the noise term should be more clearly defined.

4. Is it necessary to give the unit of amplitude error for Table1, Table3 and Table5? Because the unit of dB has been added to the analysis of amplitude error in section 2.2.

5. In the article, only two calibration receiving antennas are used in both simulation and physical experiments, is it too different from the Phased Array Antenna mentioned in the article?

6. Section 3.1 mentions “the signal processing block, which is described in Fig. 4a, comprising an AD9361 card and a ZC706 Xilinx FPGA board; a reference antenna; two calibration receiving antennas”. But in Fig. 4a, only the AD9361 card and a ZC706 Xilinx FPGA board are labeled, the location of the reference antenna and calibration receiving antenna are not labeled. fig. 4b is too blurry, need to change to a clearer picture.

7. A radiation pattern for the ideal calibration can be added to Figure 7 to show how the calibration method in this paper fits the ideal calibration.

8. In describing the principle of the autocorrelation algorithm, it is suggested to add a detailed explanation of the steps of the algorithm, especially the performance in a noisy environment. In addition, the design of the calibration system and the experimental setup are described in the experimental section, but a detailed explanation of the experimental conditions (e.g., noise level, type of interference source, etc.) is lacking.

9. The concluding section could emphasize more on the practical application value of the research and future research directions to enhance the impact of the paper.

Author Response

Comments 1: In the introduction, the authors mention the mutual coupling method [18-22], “in the case of a non-uniform arrangement of antenna elements, this method necessitates the use of a complex calibration algorithm, which in turn results in a reduction in calibration accuracy”, for the autocorrelation calibration method in this paper in the case of a non-uniform arrangement of antenna elements, this method necessitates the use of a complex calibration algorithm, which in turn results in a reduction in calibration accuracy? The authors also mentioned that the time required for calibration of near-field probes [23-27] increases significantly as the size of the array antenna increases, will the adaptive calibration proposed in this paper also be affected by this? Because the authors only studied the high precision adaptive calibration for amplitude and phase in the whole paper, they did not analyze whether the method of this paper has any effect on the uneven arrangement of the antenna elements and the variation of the size. But in the references it is mentioned that other methods can have these problems.

Response 1: Thank you for pointing this out. This paper does not address the issue of non-uniform arrangement of antenna elements. In order to enhance the precision of signal calibration in this case, it is necessary to implement a sophisticated processing algorithm, as is the case with both the mutual coupling and autocorrelation methods. It would therefore be inaccurate to present such a discussion in the introduction, which represents further work by the authors.

It should be noted that this paper presents a comprehensive investigation into the calibration of phased antenna arrays with uniform arrangement of antenna elements using an autocorrelation algorithm in the presence of noise. In addition to providing a comprehensive account of the operational principles of the proposed calibration system, we have also subjected it to rigorous experimental scrutiny under conditions characterised by the presence of internal noise and external interference. These tests were conducted using far-field scanning phased array antennas. Furthermore, the proposed method employs a single reference antenna for calibration, a feature that markedly reduces the calibration time in comparison to the near-field method, particularly when the size of the phased antenna array is considerable. In addition, the near-field method is not suitable for in-the-field calibration.

=> Introduction

Phased array antennas (PAAs) are a pervasive technology in contemporary radio engineering and communication systems [1–4]. The process of thinning PAAs involves the tuning of antenna elements with uniform spacing to achieve a desired amplitude density across the aperture area. One significant challenge in phased array signal processing is the occurrence of amplitude and phase errors across array channels [5–8]. This error results in a significant reduction in the performance of the system, particularly with respect to the estimation of the unknown steering vector for the target echo signal and the accuracy of the antenna’s weight vector. Such degradation can result in amplitude–phase distortion, namely, an amplitude–phase discontinuity of the output target echo signal. This can ultimately affect a number of performance aspects, such as the anti-jamming capability, the accuracy of the digital beamforming, and the resolution of synthetic aperture imaging in a variety of applications. In such cases, it is necessary to accurately determine the amplitude and phase shifts of the received signal between channels.

The issue has been addressed through the utilization of an array of calibration techniques, encompassing a calibration line, peripheral fixed probe, mutual coupling, near-field probe, and far-field probe. The calibration line method [9–12] permits self-calibration through the embedded lines. However, the increase in the requisite number of coupled lines and the potential for error introduced by the coupled line render this method impractical for large-scale PAAs. Additionally, the peripheral fixed-probe method [13–17] employs a probe positioned between the antennas to rectify any discrepancies. Unfortunately, the PAA must be large because of the probe inserted in the array. The method, using the near-field probe antenna [18–22], can detect the amplitude and phase of each element of the PAA through the application of mechanical movement. Still, as the antenna array size increases, the time required for calibration increases significantly because of the mechanical movement of the probe across the entire array. Furthermore, this method is not suitable for in-the-field calibration.

Conversely, the far-field calibration method [23–29] entails the measurement of the combined electric field vector of the PAA based on the reference antenna in the far-field region with the objective of calibrating the magnitude and phase of each element. This method allows for the straightforward calibration of the amplitude and phase of each element based solely on the measurement of the received power. The rotating-element electric-field vector (REV) method entails the rotation of the phase of each antenna element in the PAA from 0° to 360° to identify the maximum and minimum levels of the combined electric field vectors (REVmax and REVmin, respectively). However, a limitation of this approach is its narrow dynamic range with regard to the change in phase around the maximum and minimum received power. This results in the system being unable to accurately determine the requisite phase calibration values because of the limited power resolution near the maximum and minimum points of the combined electric field vectors, resulting in an inherent calibration error.

Furthermore, the efficacy of the calibration processes utilizing these algorithms is significantly compromised by the influence of internal noise and external interference during the operation of PAAs. The objective of a calibration method at the in-field level is to achieve high accuracy because PAAs should exhibit optimal performance and accuracy when mass-produced for use in the field to achieve good reliability. Accordingly, this study puts forth a novel far-field calibration methodology to address the shortcomings of the traditional REV techniques. The proposed method determines the phase and amplitude compensation weights subsequent to the calculation of the maximum value of the autocorrelation function (ACF). The proposed algorithm enables minimization of the impact of complex noise environments during the in-field calibration of PAAs.

The remainder of this paper is configured as follows: Section 2 presents the mathematical model and the simulation approach for the proposed method. Section 3 details the design and measurement of the proposed calibration system, followed by a comparison between the proposed method and existing REV methods. The paper is concluded in Section 4.

Comments 2: Section 2.1 mentions “We find a mathematical formula that represents the calibration process of a phased array antenna in a noisy environment”. Is the formula cited, and if so, please cite the reference to the formula.

Response 2: Thank you for pointing this out. We agree with this comment. We've already checked and corrected it.

Comments 3: In equations (1) to (7), it is recommended that all variables be defined at first occurrence, and in particular the noise term should be more clearly defined.

Response 3: Thank you for pointing this out. We agree with this comment.

=> In the following, we derive a mathematical formula that represents the calibration process of a PAA in a noisy environment.

1. Receive signals in channels of the following form:

,

(1)

where  is the complex useful signal obtained at the i-th element of the array antenna, i = 1, 2, ..., N, at time t, where  and  are the amplitude and phase of the useful signal obtained, respectively, and  is the corresponding noise, which includes internal noise and external interference.

2. Multiply the received signal  by the reference signal . The signal at the output of the multiplier has the following form:

(2)

where  and  is the complex reference signal at time t, where and are the amplitude and phase of the reference signal, respectively; and  is the corresponding noise, which exclusively pertains to the internal noise generated during the signal generation and transmission process to the multipliers. In the case of white noise, we know that the correlation function of white noise has the following form [30]: , where  and  is zero at all points except

3. Find the maximum value of the ACF  in the block for determining :

(3)

where  denotes the ACF of the received signal  and the reference signal .

4. Calculate the weight  needed to compensate for the input signal if channel k (k = 1, 2, ..., N) is taken as the standard according to the following formula:

(4)

When the kth receiver channel is the standard channel, .

5. Multiply the received signal  by the weight to be compensated :

(5)

When the kth receiver channel is the standard channel, .

6. The array coefficient function at the adder output is written as follows:

(6)

When the first receiver channel is the standard channel, the array coefficient function  output is written as follows:

(7)

Comments 4: Is it necessary to give the unit of amplitude error for Table1, Table3 and Table5? Because the unit of dB has been added to the analysis of amplitude error in section 2.2.

Response 4: Thank you for pointing this out. We agree with this comment. We've already checked and corrected it.

Comments 5: In the article, only two calibration receiving antennas are used in both simulation and physical experiments, is it too different from the Phased Array Antenna mentioned in the article?

Response 5: Thank you for pointing this out. We agree with this comment. The proposed autocorrelation calibration method was compared and analyzed with traditional rotating electric field vector methods (REVmax and REVmin) in the paper. Through actual measurement data, the author demonstrated that autocorrelation methods are more accurate than traditional methods in determining amplitude and phase offsets. In the case of an array of N elements (N > 2), the autocorrelation method continues until the final sequence is reached. Once an initial pair of elements is selected for comparison and calibration, one of the two elements is retained as the reference element for contrast with the subsequent element. This approach also serves to minimize the phase variation among the phase shifters, which is attributable to the intrinsic nature of the active component. The autocorrelation method facilitates precise determination of the amplitude and phase shifts, enabling the calibration of large-scale phased array antennas with uniform arrangement of antenna elements to achieve the maximum combined beam peak after calibration in the presence of external noise affecting performance. In contrast, the REV methods yield inaccurate amplitude and phase shifts for the initial two elements, which can result in erroneous amplitude and phase shifts for subsequent elements. This can lead to a reduction in the amplitude of the combined beam at the output combiner, which affects the efficiency of subsequent signal processing.

Comments 6: Section 3.1 mentions “the signal processing block, which is described in Fig. 4a, comprising an AD9361 card and a ZC706 Xilinx FPGA board; a reference antenna; two calibration receiving antennas”. But in Fig. 4a, only the AD9361 card and a ZC706 Xilinx FPGA board are labeled, the location of the reference antenna and calibration receiving antenna are not labeled. fig. 4b is too blurry, need to change to a clearer picture.

Response 6: Thank you for pointing this out. We agree with this comment. Therefore, a clarification has been added to the text.

=> The distance between two calibration receive antennas was ? = 0.625? and ? = 1.25?; the distance between the phased array antenna under calibration and the reference antenna was 3 m, which satisfied the far-field criteria at 3 GHz.

Figure 4. Calibration system diagram for REVmax, REVmin, and autocorrelation methods.

Figure 5. (a) Signal processing block, (b) receiving antenna, and (c) reference (transmitting) antenna.

Comments 7: A radiation pattern for the ideal calibration can be added to Figure 7 to show how the calibration method in this paper fits the ideal calibration.

Response 7: Thank you for pointing this out. We agree with this comment. A radiation pattern for the ideal calibration is added to Figure 3 in subsection 2.2.

Comments 8: In describing the principle of the autocorrelation algorithm, it is suggested to add a detailed explanation of the steps of the algorithm, especially the performance in a noisy environment. In addition, the design of the calibration system and the experimental setup are described in the experimental section, but a detailed explanation of the experimental conditions (e.g., noise level, type of interference source, etc.) is lacking.

Response 8: Thank you for pointing this out. We agree with this comment. Therefore, a clarification has been added to the text:

=> To calibrate the phase array antenna by the autocorrelation algorithm, the following steps are performed:

Step 1. Multiply the received signal with the reference signal.

Step 2. Determine the maximum value of the function at the output of the correlation multiplier.

Step 3. Caculate the weight that is required to compensate for the input signal.

Step 4. Multiply the received signal by the weight to be compensated.

Step 5. Formulate the signal that combined all the compensated signals into each channel.

The following initial parameter values were set: the distance between two calibration receive antennas was ? = 0.625? and ? = 1.25?; the distance between the phased array antenna under calibration and the reference antenna was 3 m, which satisfied the far-field criteria at 3 GHz; the distance between the reference antenna and the interference source was 2 m; the distance between the phased array antenna under calibration and the interference source antenna was 4 m; the interference power was equal to 10 dBm; and interference spectrum bandwidth was equal to 20 MHz.

More details are written in the article.

Comments 9: The concluding section could emphasize more on the practical application value of the research and future research directions to enhance the impact of the paper.

Response 9: Thank you for pointing this out. We agree with this comment.

=> The excitation of a phased array element (in terms of both amplitude and phase) in a noisy environment inevitably deviates from the ideal values, which would otherwise result in degradation of the array's performance. Therefore, careful calibration and compensation are essential to achieve the optimal results for the design of a practical phased array system. This work put forward a novel far-field calibration system with the potential to enhance accuracy and reduce system complexity.

The proposed method is distinguished from existing far-field-based solutions through the utilization of an autocorrelation algorithm. In contrast, the conventional methods based on the REV approach track the maximum and minimum magnitudes of two vector-sum elements in the array. Subsequently, the proposed method was validated through the implementation of a calibration system at 3 GHz, and its performance was benchmarked against that of conventional REV methods. To ascertain the accuracy of the proposed system, 10 trials were conducted. The simulated results obtained with the proposed method were found to align with the measurement results obtained using the far-field measurement method. A comparison of the far-field measured results revealed that the proposed method is both feasible and efficient. Therefore, the proposed method can be considered an effective solution for large-scale phase calibration in both in-field and in-factory settings, even in the presence of external noise.

It should be noted that in the case of the non-uniform arrangement of the PAA antenna elements, the proposed method requires a complex calibration algorithm, which leads to a decrease in the calibration accuracy. Therefore, the author's plans include improving the algorithm for signal calibration under interference conditions for PAAs with non-uniform arrangement of antenna elements, as well as the study of signal calibration characteristics under these conditions.

Reviewer 2 Report

Comments and Suggestions for Authors

The completeness of theoretical models and experimental verification:

This paper conducts in-depth research on the problem of calibrating phased array antennas using autocorrelation algorithms in noisy environments, and provides a mathematical model for autocorrelation calibration methods. The author not only described in detail the principle of the proposed calibration system, but also experimentally verified its performance in environments containing internal noise and external interference through far-field scanning phased array antennas. This comprehensive exploration from theory to experiment is of great significance for understanding the application of autocorrelation algorithms in phased array antenna calibration. It is suggested that the author further explore the impact of various parameters in the model on calibration accuracy in future work, and how to improve calibration efficiency through algorithm optimization.

Comparative analysis with traditional methods:

The proposed autocorrelation calibration method was compared and analyzed with traditional rotating electric field vector methods (REVmax and REVmin) in the paper. Through actual measurement data, the author demonstrated that autocorrelation methods are more accurate than traditional methods in determining amplitude and phase offsets. This comparative analysis not only enhances the persuasiveness of the paper, but also provides readers with a basis for selecting calibration methods. Suggest the author to add more details in the comparative analysis, such as experimental conditions, data processing methods, etc., so that readers can better understand and reproduce the experimental results.

Application prospects and potential improvements:

The paper points out that the proposed autocorrelation calibration method performs well in large-scale on-site and factory level phase calibration, and can even work effectively in the presence of external interference. This discovery is of great significance for the application of phased array antennas in complex environments. However, the author should also note that in practical applications, the calibration of phased array antennas may be affected by various factors, such as weather conditions, equipment aging, etc. Therefore, it is suggested that the author explore the impact of these factors on calibration accuracy in subsequent research and propose corresponding improvement measures. In addition, the author may consider extending this method to other types of antennas or communication systems to further broaden its application scope.

Author Response

Comments 1: The completeness of theoretical models and experimental verification:

This paper conducts in-depth research on the problem of calibrating phased array antennas using autocorrelation algorithms in noisy environments, and provides a mathematical model for autocorrelation calibration methods. The author not only described in detail the principle of the proposed calibration system, but also experimentally verified its performance in environments containing internal noise and external interference through far-field scanning phased array antennas. This comprehensive exploration from theory to experiment is of great significance for understanding the application of autocorrelation algorithms in phased array antenna calibration. It is suggested that the author further explore the impact of various parameters in the model on calibration accuracy in future work, and how to improve calibration efficiency through algorithm optimization.

Response 1: Thank you for pointing this out. We agree with this comment. We appreciate your advice and will take it into account in future work. In the framework of this paper, the paper added a radiation pattern for the ideal calibration and the study results of the dependence of the calibration efficiency of the phased antenna array on different azimuth angles of the transmitted antenna. More details are written in the article.

Comments 2: Comparative analysis with traditional methods:

The proposed autocorrelation calibration method was compared and analyzed with traditional rotating electric field vector methods (REVmax and REVmin) in the paper. Through actual measurement data, the author demonstrated that autocorrelation methods are more accurate than traditional methods in determining amplitude and phase offsets. This comparative analysis not only enhances the persuasiveness of the paper, but also provides readers with a basis for selecting calibration methods. Suggest the author to add more details in the comparative analysis, such as experimental conditions, data processing methods, etc., so that readers can better understand and reproduce the experimental results.

Response 2: Thank you for pointing this out. We agree with this comment.

=> To verify the reliability of the proposed method when using this calibration system, a total of 10 trials were repeated. The following initial parameter values were set: the distance between two calibration receive antennas was ? = 0.625? and ? = 1.25?; the distance between the PAA under calibration and the reference antenna was 3 m, which satisfied the far-field criteria at 3 GHz; the distance between the reference antenna and the interference source was 2 m; the distance between the PAA under calibration and the interference source antenna was 4 m; the interference power was equal to 10 dBm; and interference spectrum bandwidth was equal to 20 MHz. It is worth noting that the REVmax and REVmin approaches could also be implemented using the proposed calibration system.

More details are written in the article.

Comments 3: Application prospects and potential improvements:

The paper points out that the proposed autocorrelation calibration method performs well in large-scale on-site and factory level phase calibration, and can even work effectively in the presence of external interference. This discovery is of great significance for the application of phased array antennas in complex environments. However, the author should also note that in practical applications, the calibration of phased array antennas may be affected by various factors, such as weather conditions, equipment aging, etc. Therefore, it is suggested that the author explore the impact of these factors on calibration accuracy in subsequent research and propose corresponding improvement measures. In addition, the author may consider extending this method to other types of antennas or communication systems to further broaden its application scope.

Response 3: Thank you for pointing this out. We agree with this comment. I concur that in real-world scenarios, the calibration of phased antenna arrays can be influenced by a number of factors, including weather conditions, equipment aging, etc. These factors affect the amplitude and phase when signals are received in each channel. Therefore, this paper proposes a method to improve the accuracy of amplitude and phase shift determination for signal calibration. However, it does not consider the case of non-uniform distribution of antenna elements. Consequently, future studies should focus on developing an algorithm for signal calibration under interference conditions for PAAs with non-uniform arrangements of antenna elements, as well as on studying signal calibration characteristics under these conditions.

=> The excitation of a phased array element (in terms of both amplitude and phase) in a noisy environment inevitably deviates from the ideal values, which would otherwise result in degradation of the array's performance. Therefore, careful calibration and compensation are essential to achieve the optimal results for the design of a practical phased array system. This work put forward a novel far-field calibration system with the potential to enhance accuracy and reduce system complexity.

The proposed method is distinguished from existing far-field-based solutions through the utilization of an autocorrelation algorithm. In contrast, the conventional methods based on the REV approach track the maximum and minimum magnitudes of two vector-sum elements in the array. Subsequently, the proposed method was validated through the implementation of a calibration system at 3 GHz, and its performance was benchmarked against that of conventional REV methods. To ascertain the accuracy of the proposed system, 10 trials were conducted. The simulated results obtained with the proposed method were found to align with the measurement results obtained using the far-field measurement method. A comparison of the far-field measured results revealed that the proposed method is both feasible and efficient. Therefore, the proposed method can be considered an effective solution for large-scale phase calibration in both in-field and in-factory settings, even in the presence of external noise.

It should be noted that in the case of the non-uniform arrangement of the PAA antenna elements, the proposed method requires a complex calibration algorithm, which leads to a decrease in the calibration accuracy. Therefore, the author's plans include improving the algorithm for signal calibration under interference conditions for PAAs with non-uniform arrangement of antenna elements, as well as the study of signal calibration characteristics under these conditions.

Round 2

Reviewer 1 Report

Comments and Suggestions for Authors

1. Figure 5 is not shown in the paper, please add the full figure.

2. It is recommended that all formulas be aligned uniformly.

Reviewer 2 Report

Comments and Suggestions for Authors

Accept in present form